# Meeting in the Middle: Sociophonetic Convergence of Bad Bunny and J Balvin's Coda /s/ in Their Artistic Performance Speech

Elizabeth Naranjo Hayes

Classical and Modern Languages, Truman State University, Kirksville, MO 63501, USA; enhayes@truman.edu

**Abstract:** The artistic performance of identity by top Latin music artists can be heard on many Top-40 US radio stations, since, as of July 2023, 20% of the Billboard Hot 100 is (Spanish language) Latin music. This study aims to determine the variants found in the pronunciation of coda /s/, a robust phonetic differentiator of regional and social dialects, in the top songs versus in the spontaneous speech of the two top Latin music artists in the global market. Are Bad Bunny and J Balvin holding to the pronunciation of their respective regional variety in their *artistic performance speech* (APS, my term) or are they shifting to a different pronunciation? What motivations might cause a difference in the pronunciation of their APS and spontaneous speech? Bad Bunny and J Balvin's pronunciation of coda /s/ is analyzed in depth as sociophonetic data: their performances of songs from 2018 to 2020 that charted at the top of the Hot Latin Songs Billboard chart as well as on The Billboard Hot 100 chart, and their spontaneous speech from their most-viewed Spanish-language interviews and Instagram Live recordings on YouTube recorded between 2018 and 2020. Bad Bunny overwhelmingly used deletions (∅) in his spontaneous speech—which is typical of an island Puerto Rican—but used a statistically significant amount of maintenance of the sibilant [s] and its aspirated variant [h] in his APS ($p < 0.0001$). J Balvin primarily used [s] in his spontaneous speech—which is typical of Medellín, Colombia—but used about 50/50 [s] and (∅) in his APS. They are both shifting to a different pronunciation in their APS and converging towards each other, and the difference is statistically significant ($p < 0.0001$). This dialect convergence could be the beginning of an identity-based pan-Latinx dialect leveling that is, on the one hand, the "in-crowd" pronunciation with covert prestige but, on the other hand, is part of the formation of an evolving multi-regional connector variant diffused through popular music and pop culture.

**Keywords:** sociophonetics; sociolinguistics; language attitudes; identity; Latin music

## 1. Introduction

The next time you turn on a Top-40 radio station in the United States (US), chances are, you might hear Latin music. The Billboard Hot 100 list of top songs in the US for the week of 8 July 2023, which is compiled from the week's most popular current songs across all genres, has 20 Spanish-language songs (Billboard Charts 2023). Latin music continues to set new records in the US, but more importantly to this study, "it has been hailed as a symbol not just of important demographic trends in the United States but of an emergent pan-Latin American identity" (Marshall 2008, p. 132). This identity is reflected in the sociophonetics of the top Latin music artists, and it is evolving in an environment in the US where diverse ethnic identities are being respected and embraced by a growing segment of the population.

This mixed-methods study looks at the creation of identity via sociophonetics (coda /s/) and whether pronunciation in *artistic performance speech*[1] deviates from pronunciation in spontaneous speech. I chose the two top Latin music artists in the global market between 2018 and 2020, Bad Bunny and J Balvin, because during this time, the artists seemed parallel in their US and global audience, and they were two of the most prominent Latinx voices

with the biggest platforms. After 2020, Bad Bunny's career skyrocketed, and J Balvin took time off to enjoy fatherhood, but he is back in 2023 with a new album.

Regardless of the topics about which they are singing, they have the most potential to impact the speech of their wider audience. Coda /s/ pronunciation was chosen as it is a salient feature in the artists' APS and spontaneous speech, which are indexical of their identity. Do the two artists sing the way they speak as far as their coda /s/ pronunciation? This study aims to determine the variants found in the pronunciation of coda /s/ in the top songs versus in the spontaneous speech of the two top Latin music artists in the global market. Are they holding to the pronunciation of their respective regional variety in their APS, or are they shifting to a different pronunciation? What motivations might cause a difference in the pronunciation of their APS and spontaneous speech?

## 2. Background

### 2.1. Coda /s/ Variation in Spanish

The realization of coda /s/ in Spanish is "perhaps the single most useful parameter in dialectological descriptions" (Lipski 1984, p. 31) because it is a highly salient feature that indexes numerous social properties, and it is commonly used to situate a speaker socially (Chappell 2018). Lipski stated that "well over half of the world's Spanish speakers use dialects in which there is at least some /s/-reduction, making this process perhaps the most robust phonetic differentiator of regional and social dialects" (Lipski 2011, p. 75).

The greatest variation in the pronunciation of consonants in Spanish occurs in coda, syllable-final position, because it is universally regarded as the weakest in terms of neutralization or total effacement (Lipski 2012, p. 5). The most common modification of Spanish coda consonants involves /s/ (Lipski 2011, 2012). In addition to the voiceless alveolar fricative [s] with high-frequency sibilance, Hualde (2014) described two distinguishable /s/ allophones based on weakening that occurs in free variation; the pronunciation can vary by the same speaker in the same phonological context depending on formality:

- Aspiration—[h] glottal frication
- Deletion—[Ø] total deletion of the sibilant

(Hualde 2014, pp. 12–13, 157; see also Schmidt 2012, p. 190)

#### 2.1.1. Coda /s/ in Puerto Rican Spanish

Bad Bunny, one of the two artists whose speech was analyzed, is from Puerto Rico. Puerto Rican Spanish speakers can retain, aspirate, or delete the syllable-final /s/, although aspiration and deletion are widespread throughout the island and distributed across social classes and groups (Boomershine 2006; Chappell 2018).

In Puerto Rican Spanish, retention of the sibilant [s] is primarily associated with the upper socioeconomic classes and cautious or formal speech. Terrell (1977) noted that there is some evidence that the use of aspiration and deletion may be more favored by Puerto Rican males, younger generations, and lower socioeconomic classes, but that neither aspiration nor deletion of /s/ appear to be stigmatized, and both are used by all social classes of Puerto Ricans on almost all occasions. Terrell states that aspiration and deletion are regular features of the "standard" cultured speech of educated Puerto Ricans, as well as those of other socioeconomic levels, "and not a matter of 'sloppy' pronunciation" (Terrell 1977, p. 37; see also Mohamed and Muntendam 2020, p. 392). Because coda /s/ realizations such as aspiration and deletion are also used by the cultured, educated classes of Puerto Ricans, it is possible that aspiration and deletion are used as markers of Puerto Rican identity, regardless of class affiliation.

Another variant of /s/ has been noted in Puerto Rican Spanish: the use of a glottal stop [?] (Lipski 2011; Mohamed and Muntendam 2020; Terrell 1977; Valentín-Márquez 2006). Some linguistic and social factors of glottal stops tie into reggaeton (Mohamed and Muntendam 2020; Valentín-Márquez 2006). The reggaeton genre of music has roots in Jamaica; it led to Spanish Reggae, and when it reached Puerto Rico and combined with rap rhythms, it eventually became reggaeton music (Ellis 2018; Merriam-Webster n.d.). The present

study used reggaeton music as part of the corpus, so glottal stops were coded for and were considered as a variant of coda /s/ realizations. Not all variants are equally salient, especially considering the recording, signal processing, mixing, and mastering processes of the audio; thus, very minimal tokens of glottal stops were detected in the data and were, therefore, excluded from the study.

### 2.1.2. Coda /s/ in Colombian Spanish

J Balvin, the other artist whose speech was analyzed, is from Colombia, a country whose Spanish was called one of many contrasts and contradictions (Lipski 1996). The Spanish of Colombia has been generally divided by its four geographic regions (Hualde 2014; Lipski 1996): the Caribbean coast, the Pacific coast, the Amazonian region, and the interior highlands, which includes the prestige variety of the capital, Bogotá, and of Medellín, as shown in Figure 1.

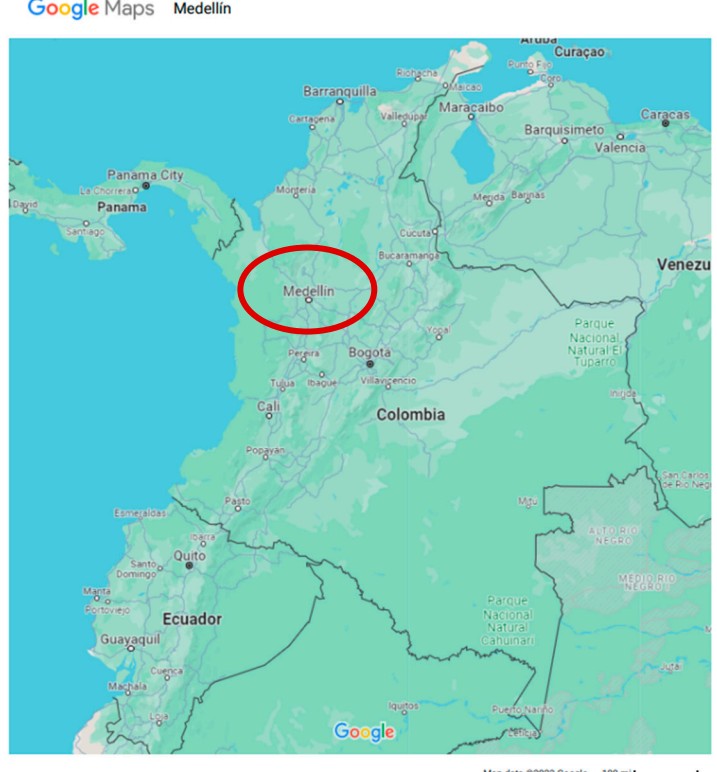

**Figure 1.** Map of Colombia, with Medellín circled in red. Source: Google Maps, 2023.

The region most relevant to this study includes Medellín, the capital of the department of Antioquia, where the present study's Colombian artist is from. Montes Giraldo (1982) placed Medellín as part of the Central Andes superdialect, which is a western Andes dialect. According to Lipski (1996), the educated speech of Bogotá and nearby cities of the interior of Colombia enjoys the popular reputation of speaking the "purest" Spanish in the Spanish-speaking world (p. 227). The speech of Bogotá possibly influences that of Medellín via the Bogotá-based norm in Colombian media, though I was unable to locate any published empirical studies. Several studies on the speech of Cali, which is further south, have documented that there is /s/ reduction occurring in both syllable initial and syllable-final positions (Brown and Brown 2012; File-Muriel and Brown 2011).

### 2.2. Indexicality

Reed (2020) explained that "often, a way of speaking is associated or recognized as being indicative of where someone is from, that is, it indexes (or points out) place" (Reed 2020, p. 2). Entire languages can become registers that index social categories, such

as the use of Spanish in the United States being indexical of US Latinx pan-ethnicity (Rosa 2010) or the use of Spanish–English codemixing being indexical of in-group Latinx identity, especially in instances where that identity is not readily discernable.

In the Appalachian region of the US, a feature that seems to have social meaning and has come to index certain aspects of regional and local identity is "the monophthongization of /aɪ/, e.g., *I*, *ride*, *right*, *realized* as [aː], [ɹaːd], and [ɹaːt] respectively" (Reed 2014, p. 159). Similarly, Preston (2018) gives an example of a cartoon in which two men explain how the term "NASCAR" (which has an /æ/ vowel) was invented: one points to a car and says "NAS CAR, HUH?" and the other replies, "YUP. REAL NAS!" The implication is they are from the American South due to the monophthongization of /aɪ/ (Preston 2018, p. 48).

Mack (2011) discusses the topic of socioindexicality, that certain patterns of pronunciation can index social information. She explains that listeners can make judgements on the social variables of the speaker, such as their gender, age, country/region of origin, educational level, socioeconomic status, ethnicity, and sexual orientation (Mack 2011, p. 81). Mack found that in Puerto Rican Spanish, there is a possible connection between the perceived sexual orientation of a speaker and the speaker's realization of syllable-final /s/, where there was a strong association between the sibilant [s] variant and the perception of the male speaker as "sounding gay" (p. 90). Rosa (2010) gives an example of indexing: "Familiar Caribbean phonological features of both his heavily accented English and native Spanish (e.g., syllable-final /s/ deletion/aspiration, intervocalic/word final /d/ deletion, etc.) . . . suggested to me that he was Puerto Rican" (p. 194).

### 2.3. Language Attitudes

Negative language attitudes toward the pronunciation and language represented in reggaeton music abound on social media. The Twitter post in Figure 2 went viral and sparked many memes: "Más de dos errores ortográficos en un párrafo ya es reggaetón". 'More than two spelling errors in a paragraph, now it's a reggaeton' (Aristizábal 2017).

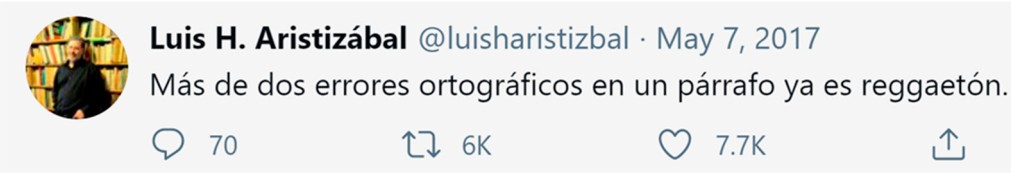

**Figure 2.** Tweet that led to memes. Source: (Aristizábal 2017).

La Real Academia Española (rae.es), which is the academic ruling body over the Spanish language, has also commented on the term *bebesita* 'baby girl' that was used by the former reggaeton power couple Karol G and Anuel AA, saying that it should be spelled *bebecita*, although it is not a word that is found in its dictionary (Judith 2019). The Royal Spanish Academy also tweeted that the proper Spanish spelling of the popular Latin music genre is *reguetón* (Real Academia Española 2019) and recently added it to its dictionary (Real Academia Española 2021). Despite that, Iraís (2020) stated that most Latin music artists are adamant about keeping the original Anglicized spelling: reggaeton.

### 2.4. Covert Prestige

Aspiration and deletion of coda /s/ are viewed as non-normative speech behavior, and deletion is likely the most stigmatized of coda /s/ variants (Erker and Otheguy 2016, p. 135). The Latin music artist of Mexican ancestry from Los Angeles, California, Becky G, has been accused of "using a Caribbean Blaccent" in her music and her APS, as reported by the popular Latinx online magazines *Mitú* and *Remezcla* (Barron 2019; Caraballo 2019). Social media comments concurring with this position were posted by Latinx, such as the following tweet:

> #HowDoMexicansTalk You are lying if you're going to deny non-Black, non-Caribbean artists/people are increasingly adopting Caribbean and African Amer-

ican ways of speaking to market themselves in Black/Afro-Latinx genres and culture, this has been peeped by Black Latinxs for YRS. (Ureña-Ravelo 2019)

Becky G published a response article to the criticisms in *PopSugar Latina* magazine, stating that she may not speak Spanish perfectly but that the lack of language knowledge does not take away from her Latinaness (Becky G 2017). She also posted a multipart video response on Instagram with her mother, defending the way she speaks (Becky G 2018).

She asserts that any performance of identity that is indexical of Caribbean Spanish is not intentional, although social media commentary to the contrary abounds. The huge backlash from the Latinx community is specifically from people who grew up near her hometown or knew her and can attest to her pronunciation in her APS as being fake:

> Grew up near Becky G's hometown & her fake Caribbean accent is NOT #HowDoMexicansTalk. Stans are disingenuous and the Afro-Latinx person who called Becky out was correct. . . (Rubén Angel 2019)

To shed some light on Becky G's situation, we will consider Magro's (2016) study. He conducted a study on language attitudes of Dominican students at a university in New York City regarding their variety of Spanish compared to the Spanish imposed in formal or academic settings. Magro used a matched-guise technique with rap music recorded with Dominican indexes and with standard Peninsular indexes, followed by interviews with the participants. He concluded that when it comes to linguistic ideologies, the standard Peninsular is considered the authentic "real" Spanish, as evidenced by its use for pedagogy. This creates linguistic insecurity in the speakers of the stigmatized varieties and can lead to them abandoning their regional variant. On the contrary, in the field of hip-hop in Spanish, "the stigmatized way of talking of Dominican heritage speakers becomes the prestige variety, the proper one in this context" (Magro 2016, p. 24). Magro explained that the covert prestige of stigmatized varieties in hip-hop and rap is due to the authenticity and credibility of the language as it relates to the streets. This might be the authenticity and credibility that is being performed by artists such as Becky G during their APS.

## 3. Methodology

### 3.1. Participants: Bad Bunny and J Balvin

This study focuses on the publicly accessible recordings of the APS and spontaneous speech of the two top Latin music artists from 2018 to 2020, per Billboard Charts: Bad Bunny and J Balvin.

The two artists chosen for this study are the highest grossing, with the most songs on both the Hot Latin Songs Billboard Chart and on The Hot 100 Billboard chart, as well as having the most social media followers (2018–2020): Bad Bunny, from Puerto Rico, and J Balvin, from Colombia. They can be viewed as examples in a microcosm of a larger dynamic, in that neither was raised within the US-based Latinx context but both have strong connections to the US within their backgrounds, and both operate in the arena of popular music where the US dominates. They are part of a global Spanish-speaking community, so their linguistic practices may serve to illustrate a complex dynamic: the use of sociophonetic features that are emblematic of the larger global Spanish-speaking world that they identify with and which is their audience.

### 3.1.1. Biography of Bad Bunny

Benito Antonio Martínez Ocasio was born on 10 March 1994, in Vega Baja, Puerto Rico—a municipality located on the coast of north central Puerto Rico (González 2020). He grew up in a Catholic, two-parent, lower-middle-class family, with two younger brothers in what he considers the countryside, with maybe 4 trips a year to San Juan (del Valle Schorske 2020). He was working as a bagger at an Econo supermarket and making music while he studied audiovisual communications at the University of Puerto Rico at Arecibo, but he only completed his first two years of college before he was discovered on

SoundCloud.com and signed to a record label (del Valle Schorske 2020; El Comercio 2019; González 2020; Milenio 2019).

Bad Bunny has numerous singles and song collaborations as well as albums after 2020, which are not included in this study. Bad Bunny's Chart History in Table 1 and his social media followers in Table 2 show the overwhelming platform Bad Bunny wields in his personal and professional online presence, as well as the broad scope of his impact on his followership.

**Table 1.** Bad Bunny Chart History as of 10 July 2023.

| Billboard Chart | The Hot 100 | Hot Latin Songs |
|---|---|---|
| Songs per Category | 1–Number 1<br>10–Top 10<br>70–Total | 12–Number 1<br>61–Top 10<br>147–Total |

**Table 2.** Bad Bunny social media followers as of 10 July 2023.

| Social Media Platform | Twitter | Instagram | Facebook | YouTube |
|---|---|---|---|---|
| Handle | @sanbenito | @badbunnypr | @badbunnyofficial | YouTube.com/c/badbunnypr/ (accessed on 10 July 2023) |
| Followers | 4.9 million Followers | 45.8 million Followers | 14 million Followers | 46.1 million Subscribers<br>30.7 billion Views |

### 3.1.2. Biography of J Balvin

Jose Álvaro Osorio Balvin, who pronounces his first name with the stress on the first syllable, without an accent on the final é, was born on 7 May 1985, in Medellín, Colombia. His father was a well-to-do businessman, although, at the time, Medellín was a violent, dangerous place (Collins 2019; Galvez 2020). He listened to Kurt Cobain, Tupac, Biggie, Nirvana, and Metallica growing up, along with the Spanish-language music of his parents (Borge and Castillo 2020; Collins 2019; J Balvin 2018). When J Balvin was 17, he participated in a foreign-exchange program to Oklahoma to learn American English; then, he returned to Colombia (IMDb n.d.). J Balvin started freestyle rapping as a youth, and he was a national freestyle rap champion of Colombia for three consecutive years (J Balvin 2018). He returned to the US, this time to Staten Island, New York, and worked odd jobs, such as a dog walker, house painter, and roofer, but he again returned to Colombia, this time to study international business at the University of Medellín, as well as to start making music under the name J Balvin (IMDb n.d.). He left school after 3 years, because YouTube helped launch his music internationally (J Balvin 2018).

J Balvin has numerous singles, song collaborations, and albums before 2020, which are not included in this study, and after 2020, he took time off to enjoy fatherhood. He is back in 2023 with new music. J Balvin's Chart History in Table 3 and his social media followers in Table 4 show the enormous platform J Balvin wields in his personal and professional online presence, as well as the broad scope of his impact on his followership.

**Table 3.** J Balvin Chart History as of 10 July 2023.

| Billboard Chart | The Hot 100 | Hot Latin Songs |
|---|---|---|
| Songs per Category | 1–Number 1<br>2–Top 10<br>18–Total | 9–Number 1<br>35–Top 10<br>93–Total |

**Table 4.** J Balvin social media followers as of 10 July 2023.

| Social Media Platform | Twitter | Instagram | Facebook | YouTube |
|---|---|---|---|---|
| Handle | @jbalvin | @jbalvin | @jbalvinoficial | YouTube.com/c/jbalvin/ (accessed on 10 July 2023) |
| Followers | 10.7 million Followers | 51.7 million Followers | 26 million Followers | 34.1 million Subscribers 23 billion Views |

*3.2. Data Collection*

3.2.1. Corpus of Artistic Performance Speech

The songs that were used to create a corpus of APS are songs that peaked in the top 10 on the Hot Latin Songs Billboard chart and which also charted on The Hot 100 Billboard chart from 2018 to 2020. Though many of the songs feature several artists, only the tokens by Bad Bunny and J Balvin were considered. The songs and their official videos are all available on my website: https://enhayes.people.ua.edu/ (accessed on 10 July 2023). The songs are presented in three groups in Table 5; five are Bad Bunny and J Balvin collaborations, six include Bad Bunny, and six include J Balvin.

**Table 5.** APS by artist(s), song, year, duration, and number of coda /s/ tokens.

| **Bad Bunny and J Balvin Collaborations** |
|---|
| Cardi B, Bad Bunny, J Balvin—"I Like It"—2018—4:18—36 tokens |
| J Balvin, Bad Bunny—"La canción"—2019—4:11—83 tokens |
| J. Balvin, Bad Bunny—"Qué pretendes"—2019—3:44—82 tokens |
| Jhay Cortez, J. Balvin, Bad Bunny—"No me conoce (Remix)"—2019—5:06—48 tokens |
| J. Balvin, Dua Lipa, Bad Bunny, Tainy—"Un día (One Day)"—2020—4:10—38 tokens |
| **Bad Bunny** |
| Casper, Nio García, Darell, Nicky Jam, Bad Bunny, Ozuna—"Te boté, Remix"—2018—7:03—17 tokens |
| Bad Bunny, Drake—"Mía"—2018—3:31—59 tokens |
| Bad Bunny—"Vete"—2019—3:28—83 tokens |
| Bad Bunny—"Callaíta"—2019—4:12—38 tokens |
| Bad Bunny, Jhay Cortez—"Dákiti"—2020—3:34—63 tokens |
| Bad Bunny—"Si veo a tu mamá"—2020—2:51—30 tokens |
| **J Balvin** |
| Nicky Jam, J. Balvin—"X (Equis)"—2018—3:12—26 tokens |
| Anuel AA, Daddy Yankee, Karol G, Ozuna, J Balvin—"China"—2019—5:02—17 tokens |
| Ozuna—"Baila baila baila (Remix)" Ft. Daddy Yankee, J Balvin, Farruko, Anuel AA—2019—3:57—14 tokens |
| Black Eyed Peas, J Balvin—"Ritmo (Bad Boys for Life)"—2020—3:39—38 tokens |
| DJ Snake, J. Balvin, Tyga—"Loco contigo"—2020—3:10—39 tokens |
| Sech, Daddy Yankee, J Balvin, Rosalía, Farruko—"Relación, Remix"—2020—4:10—25 tokens |

3.2.2. Corpus of Spontaneous Speech

The videos chosen were the most-viewed, unscripted, informal conversations in Spanish of at least five-minute duration by Bad Bunny and J Balvin from 2018 to 2020, as searched on YouTube.com (accessed on 29 November 2023). They are presented in three

groups in Table 6; one is a Bad Bunny and J Balvin collaboration interview, three are of Bad Bunny, and three are of J Balvin.

**Table 6.** Spontaneous speech by artist(s), interview, year, duration, and number of coda /s/ tokens.

| **Bad Bunny and J Balvin Collaboration Interview** |
| --- |
| Bad Bunny and J Balvin Talk Upcoming Joint Album and the Rise of Latin Trap—*Complex Cover*—2018—31:58—881 tokens |
| **Bad Bunny** |
| Bad Bunny nos habla de cómo el éxito le cambió la vida—*Don Francisco Te Invita*—2018—11:48—273 tokens |
| Bad Bunny Goes *Sneaker Shopping with Complex*—2018—9:30—272 tokens |
| Bad Bunny—*YouTube Artist Spotlight Stories*—2020—10:27—199 tokens |
| **J Balvin** |
| J Balvin habla de sus malentendidos con Anuel AA y Maluma—*Al Grano con El Guru*—2019—27:59—571 tokens |
| J Balvin y Úrsula Corberó (La Casa de Papel)—*Instagram Live*—2020—16:45—145 tokens |
| Jugando con J Balvin a Fortnite—*TheGrefg*—2020—14:17—73 tokens |

### 3.3. Data Analysis

The official music video for each song was downloaded from YouTube Premium as a WAV file, and the song lyrics were downloaded from Google search, cross checked with Genius.com, and edited for accuracy, when necessary, in a Word file. The song lyrics were divided by verse, and time stamps were added to each song verse based on the official YouTube video. The lyrics were inspected, and all tokens of coda /s/ were identified. Like Gibson (2019), the WAV files for the songs were then uploaded to Praat (Boersema and Weenink 2023), a text grid was created, and lyrics were annotated in the text grid. Like Caillol and Ferragne (2019), source words containing the coda /s/ token were annotated and segmented in another text grid tier, and the token itself was annotated for measurements in its own text grid tier. Praat's waveforms and spectrograms were used to check for the presence or absence of frication and aided in the segmentation of source words into phonemes, as well as in the identification of phonemes. An extensive Excel file was used for coding the independent variables relevant to the hypotheses of interest. All tokens by Bad Bunny and J Balvin in all the songs in the corpus were included.

The interviews underwent a similar process: each interview was downloaded from YouTube Premium as a WAV file, and if available, the transcript was downloaded to a Word file. To aid in creating the transcription, the playback speed on YouTube (under settings) was decreased to 0.75 or 0.5 at times. The transcription was divided by phrases, based on the natural pauses in speech, and time stamps were added to each phrase based on the YouTube video. The transcription was inspected, and all tokens of coda /s/ were identified, annotated, and coded like the lyrics.

### 3.4. Categorical Labeling of Coda /s/ Variants

Coding for categorical labeling was adapted from Hualde's (2014) description of /s/ allophones, with frication noted in the spectrogram.

- Maintenance: sibilant—a voiceless laminal alveolar fricative [s]—normative Latin American pronunciation (Hualde 2014, pp. 147, 150), and its voiced sibilant allophone, which can occur in complementary distribution [z] (pp. 11–12, 154–55). These sounds produce a turbulent airstream that is visible on the spectrogram as dark static.
- Aspiration: [h] a voiceless glottal fricative (Hualde 2014, p. 157), and any instances voiced aspiration [ɦ]. These sounds appear on the spectrogram with very weak turbulence, though they are admittedly the hardest to define.

- Deletion: [Ø] total deletion of /s/ (Hualde 2014, p. 157). See Figures 3 and 4 for examples of Bad Bunny's deletions, where the sound is not present on the spectrogram.

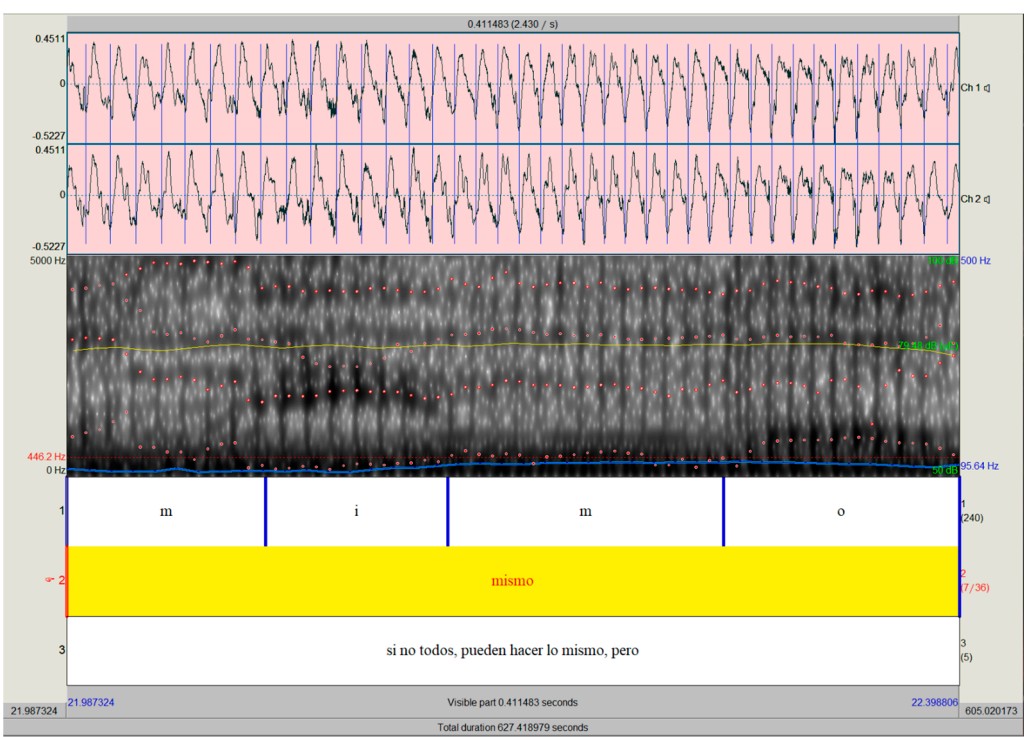

**Figure 3.** Bad Bunny, <mismo> [ˈmi.mo], showing a deletion, from YouTube interview.

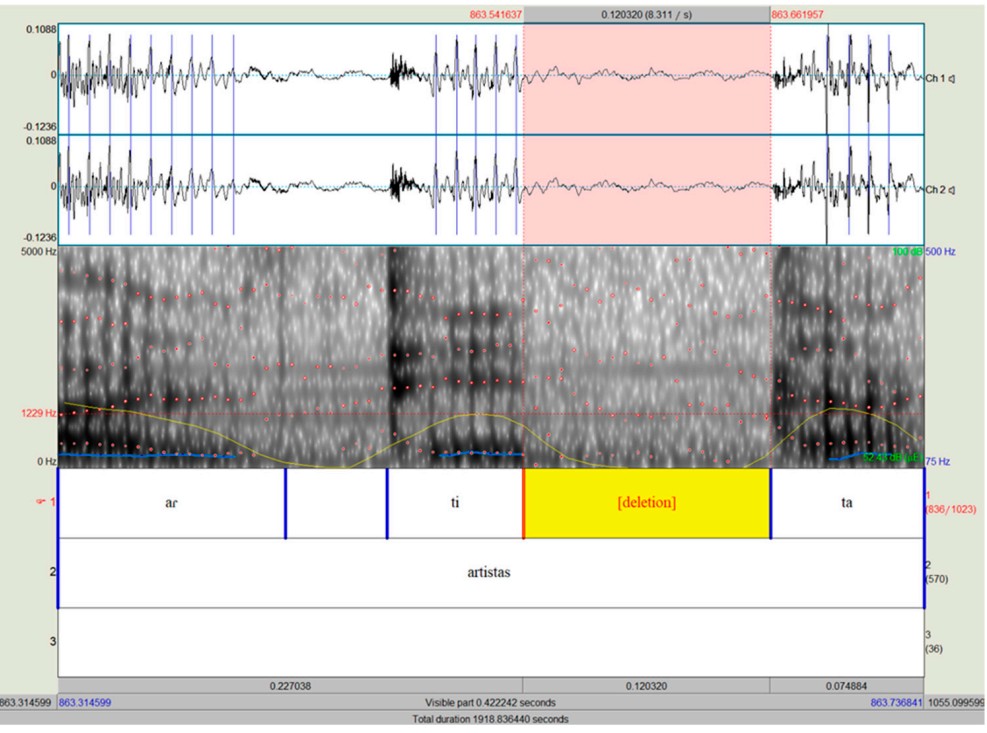

**Figure 4.** Bad Bunny, <artistas> [aɾ.ˈti.ta], showing two deletions, from YouTube interview.

*3.5. Dependent and Independent Variables*

Realization of the coda /s/ was the dependent variable, and it was further broken down into maintenance, aspiration, or deletion. Independent variables were Artist (Bad

Bunny or J Balvin) and Performance Mode (Speech or APS). Sociolinguistic factors, such as the speech of the interlocutor(s), the intended audience, and the level of formality versus informality of the conversation, are considered in the discussion as part of the qualitative analysis. A multinomial logistic regression for Artist, Performance Mode, and Realization was conducted on R to determine whether the terms were statistically significant. All analyses were performed using R Statistical Software (v4.3.2; R Core Team 2023). The multinomial logistic regression "multinom" command was used via the nnet R package (Venables and Ripley 2002). The estimated values in the tables and graphs were considered using the emmeans R package (v1.8.5; Lenth 2023).

Due to data collection being based on commercial recordings sourced from YouTube, there is information that is typically lost that can bias center-of-gravity (COG) measurements. In the original data collection, COG was used as a measurement to verify the reliability of the categorical dependent variables. It was brought to my attention that digital signal processing during the mixing and mastering of the studio recordings could impact the COG; therefore, I did not use this measurement in the final version of this study.

## 4. Results and Discussion

### 4.1. What Variants Are Found in the Pronunciation of Coda /s/ in the APS versus in the Spontaneous Speech of the Two Top Global Music Artists Bad Bunny and J Balvin?

4.1.1. APS

The variants found in the overall APS are seen in Table 7. Bad Bunny shows a higher preference for aspiration (43.31%), and J Balvin has an almost 50/50 split between maintenance and deletion.

**Table 7.** Realization of coda /s/ by Artist in APS.

| Realization of Coda /s/ | Bad Bunny-APS | | J Balvin-APS | | |
| --- | --- | --- | --- | --- | --- |
| | N | % | N | % | Total |
| Maintenance | 94 | 19.67% | **151** | **50%** | 245 |
| Aspiration | **207** | **43.31%** | 4 | 1.32% | 211 |
| Deletion | 177 | 37.03% | 147 | 48.68% | 324 |
| | 478 | 100% | 302 | 100% | 780 |

The highest number and percentage of APS realizations per artist are bolded.

When we consider the artistic performance speech (APS) of each artist, we start to see a difference in trends. Table A1 (see Appendix A) considers the Realization of coda /s/ by Artist in the APS by song. Although, overall, Bad Bunny prefers aspiration to deletion, and J Balvin prefers 50/50 maintenance and deletion, when we look further, there are several examples where Bad Bunny has an almost even split between maintenance, aspiration, and deletion. In "Qué pretendes" with J Balvin, he has 37%/37% maintenance and deletion, and in "Si veo a tu mamá", he also has very balanced percentages. These songs had enormous followership, so his pronunciation seems intentionally adjusted to allow for more syllable-final -s. Due to negative language attitudes, a more normative pronunciation may have been desired for the songs that would reach a larger audience, such as "Mía" with the Canadian artist, Drake, which has 29% maintenance. Two of the collaborative songs with J Balvin, "Un día", and "No me conoce", have deletion rates that are a bit higher than the rest, showing both indexicality and covert prestige. In these two songs, J Balvin also has slightly higher maintenance rates than in his overall APS, so in these songs, they stay the truest to their spontaneous speech realization rates.

J Balvin has an overall balance between maintenance and deletion rates in his APS, but his APS in several songs has high deletion rates, especially when performing with an international or English-speaking artist. In "I Like It" with Bad Bunny and Cardi B, which hit number one on the Billboard Hot 100, he has 93% deletions, and in "Ritmo" with the Black Eyed Peas, he has 82% deletions. This pronunciation shows a preference for the covert prestige of the Caribbean pronunciation that is in vogue in Latin music, intentionally

done for the establishment of identity. His APS pronunciation could be an alignment with the coastal pronunciation of Colombia that has higher deletion rates, or he is incorporating it stylistically to identify with it, though it is not his typical speech.

### 4.1.2. Spontaneous Speech

The variants found in the overall spontaneous speech are seen in Table 8. Bad Bunny leans heavily on deletion (93.7%), whereas J Balvin prefers maintaining sibilants (80.84%).

**Table 8.** Realization of Coda /s/ by Artist in spontaneous speech.

|  | Bad Bunny-Speech | | J Balvin-Speech | | |
| --- | --- | --- | --- | --- | --- |
| **Realization of Coda /s/** | **N** | **%** | **N** | **%** | **Total** |
| Maintenance | 44 | 3.80% | **979** | **80.84%** | 1023 |
| Aspiration | 29 | 2.50% | 91 | 7.51% | 120 |
| Deletion | **1085** | **93.70%** | 141 | 11.64% | 1226 |
| | 1158 | 100.00% | 1211 | 100.00% | 2369 |

The highest number and percentage of Speech realizations per artist are bolded.

When considering the Realization of coda /s/ by Artist in Speech by interview, in Table 9, the interplay of the interlocuter, the intended audience, and the formality of the conversation offers insight into the realization. Bad Bunny has a consistent rate of around ninety percent deletions, but the percentage is higher when he has an interlocuter who also uses deletions. In his *Complex Cover* interview, he is speaking informally with his good friend and collaborator, J Balvin, and a Puerto Rican interviewer who has high deletion rates. He and the interviewer are having a few drinks during the interview, and the longer the session goes on, the more deletions occur, which is to be expected with informal speech by speakers of his regional variant.

Bad Bunny's highest percentage of deletions is on *Don Francisco Te Invita*, speaking with the Chilean host who shows strong /s/ reduction, and in front of a huge Spanish-speaking audience. The interview took place in 2018, as Bad Bunny was beginning his career, so he used the indexicality of his pronunciation to establish a strong Puerto Rican identity as he addressed a Latino audience. Bad Bunny's lowest percentage of deletions in speech, though still high (86%), is for a global audience—a *YouTube Artist Spotlight*. There is no visible interlocuter, and he speaks in Spanish with English subtitles to allow for a larger English-speaking audience to be introduced to him. In 2020, he was still climbing the charts and appeared timid compared to the confident megastar of 2023. His speech is more intentional, and the ambiance seems more formal, which could account for higher percentages of aspiration (8%) and maintenance (6%) than in the other speech samples.

J Balvin's speech has a consistent eighty percent maintenance, with about fifteen percent deletion, and minimal aspiration. We see this trend when he is speaking very informally with a Spanish actress whom he is talking to on *Instagram Live*, when he is playing video games with *TheGrefg*, another Spaniard, and even during a longer interview with *El Guru*, a Puerto Rican that addresses some serious topics. These interlocuters are all about his age and have /s/ reducing regional variants of Spanish, but he holds to his speech rates. The interesting case for him was his *Complex Cover* interview alongside Bad Bunny. J Balvin does not drink but enjoys drinking water and conversing at length with two /s/ reducing Puerto Ricans. In this interview, there is slight speech accommodation because his maintenance dips into the seventies (76%), and his aspiration is much higher (15%) than usual for him.

**Table 9.** Realization by Artist(s) in Speech by interview.

| Speech | Realization | Bad Bunny | | J Balvin | | Totals |
|---|---|---|---|---|---|---|
| Bad Bunny and J Balvin Talk Upcoming Joint Album and the Rise of Latin Trap—*Complex Cover* | Maintenance | 14 | 3% | **322** | **76%** | 336 |
| | Aspiration | 11 | 2% | 65 | 15% | 76 |
| | Deletion | **431** | **95%** | 37 | 9% | 468 |
| | Totals | 456 | 100% | 424 | 100% | 880 |

| Speech | Realization | Bad Bunny | |
|---|---|---|---|
| Bad Bunny nos habla de cómo el éxito le cambió la vida—*Don Francisco Te Invita* | Maintenance | 7 | 3% |
| | Aspiration | 2 | 1% |
| | Deletion | **264** | **97%** |
| | TOTALS | 273 | 100% |
| Bad Bunny Goes *Sneaker Shopping with Complex* | Maintenance | 15 | 6% |
| | Aspiration | 11 | 4% |
| | Deletion | **246** | **90%** |
| | TOTALS | 272 | 100% |
| Bad Bunny—*YouTube Artist Spotlight Stories* | Maintenance | 12 | 6% |
| | Aspiration | 15 | 8% |
| | Deletion | **172** | **86%** |
| | TOTALS | 199 | 100% |

| Speech | Realization | J Balvin | |
|---|---|---|---|
| J Balvin habla de sus malentendidos con Anuel AA y Maluma—*Al Grano con El Guru* | Maintenance | **479** | **84%** |
| | Aspiration | 22 | 4% |
| | Deletion | 70 | 12% |
| | TOTALS | 571 | 100% |
| J Balvin y Úrsula Corberó—*Instagram Live* | Maintenance | **119** | **82%** |
| | Aspiration | 4 | 3% |
| | Deletion | 22 | 15% |
| | TOTALS | 145 | 100% |
| Jugando con J Balvin a Fortnite—*TheGrefg* | Maintenance | **60** | **82%** |
| | Aspiration | 1 | 1% |
| | Deletion | 12 | 16% |
| | TOTALS | 73 | 100% |

The highest number and percentage of Speech realizations per artist in each section are bolded.

## 4.2. Are Bad Bunny and J Balvin Holding to the Pronunciation of Their Respective Regional Variety in Their APS, or Are They Shifting to a Different Pronunciation?

There is a clear change in the percentage of overall deletions that Bad Bunny uses in his speech to the much higher percentage of aspiration he uses in his APS, per the tokens analyzed in this study (see Table 10). His speech is typical of an island Puerto Rican, with high rates of /s/ deletion and a bit of aspiration, and it is a shibboleth of identity, but his APS is a bit different.

**Table 10.** Changes in Realization of coda /s/ by Bad Bunny in Speech versus APS.

| Realization of Coda /s/ | Bad Bunny-Speech | | Bad Bunny-APS | | |
| | N | % | N | % | CHANGE |
|---|---|---|---|---|---|
| Maintenance | 44 | 3.8% | 94 | 19.67% | ↑ 15.87% |
| Aspiration | 29 | 2.5% | **207** | **43.31%** | ↑ 40.81% |
| Deletion | **1085** | **93.7%** | 177 | 37.03% | ↓ 56.67% |
| | 1158 | 100.00% | 478 | 100.00% | |

The highest number and percentage of Speech and APS realizations are bolded. Arrows show direction of change, whether it is an increase or a decrease.

There is a definite difference in the percentage of overall maintenance that J Balvin uses in his speech to the very balanced percentage of deletion and maintenance he uses in his APS, per the tokens analyzed in this study (see Table 11). His speech is typical of a Colombian of the central region, with maintenance but also some aspiration and deletions. His APS seems to lose aspiration and maximize on deletions, which is a change.

**Table 11.** Changes in Realization of coda /s/ by J Balvin in Speech versus APS.

| Realization of Coda /s/ | J Balvin-Speech | | J Balvin-APS | | |
| | N | % | N | % | CHANGE |
|---|---|---|---|---|---|
| Maintenance | **979** | **80.8%** | **151** | **50%** | ↓ 30.8% |
| Aspiration | 91 | 7.5% | 4 | 1.32% | ↓ 6.18% |
| Deletion | 141 | 11.6% | **147** | **48.68%** | ↑ 37.08% |
| | 1211 | 100% | 302 | 100% | |

The highest number and percentage of Speech and APS realizations are bolded. Arrows show direction of change, whether it is an increase or a decrease.

A multinomial logistic regression model was created with Realization as the response variable, and Artist and Performance Mode as predictors. Because the hypothesis involves the possibility that different artists will adjust pronunciation differently when moving between Speech and APS, interaction between Artist and Performance Mode was included in the model. A series of likelihood ratio tests of the full model versus simpler models (see Table 12) show that each term is statistically significant ($p < 0.0001$ in all cases).

**Table 12.** Sequential comparison of models from the intercept-only model to the full interaction model using the likelihood ratio test.

| Model | Test | Df | LR Stat. | Pr(Chi) |
|---|---|---|---|---|
| 1 | | NA | NA | NA |
| Artist | 1 vs. 2 | 2 | 1602.9787 | 0.0001 |
| Artist + Performance Mode | 2 vs. 3 | 2 | 240.4104 | 0.0001 |
| Artist ∗ Performance Mode | 3 vs. 4 | 2 | 555.7885 | 0.0001 |

Model 1 is the intercept model, but the four different models sequentially show how there is a statistically significant interaction among and between the variables. Artist and Performance Mode both predict percentages of the realizations, and there is an interaction both among and between them. These interactions are all statistically significant ($p < 0.0001$).

Table 13 shows model-estimated probabilities for each Realization, Artist, and Performance Mode, along with confidence intervals. Confidence interval widths were adjusted for multiple comparison using Šidák's method, within each three-estimate group. The probability that Bad Bunny will use deletion in his speech, according to the model, is 93.7% (91.6–95.8%). The probability he will use aspiration in his APS is 43.3% (36.5–50.1%). These align almost perfectly with the data in Table 10. The probability that J Balvin will use

maintenance in his speech, according to the model, is 80.8% (77.4–84.2%). The probability he will use deletion in his APS is 48.7% (40–57.3%). These align almost perfectly with the data in Table 11.

**Table 13.** Estimated Realization probabilities for each artist and performance mode, along with 95% simultaneous confidence intervals, adjusted for three estimates.

| Realization | Artist | Performance Mode | Prob | St. Error | df | Conf. Low | Conf. High |
|---|---|---|---|---|---|---|---|
| Maintenance | Bad Bunny | Speech | 0.038 | 0.006 | 8 | 0.021 | 0.055 |
| Aspiration | Bad Bunny | Speech | 0.025 | 0.005 | 8 | 0.011 | 0.039 |
| Deletion | Bad Bunny | Speech | 0.937 | 0.007 | 8 | 0.916 | 0.958 |
| Maintenance | Bad Bunny | APS | 0.197 | 0.018 | 8 | 0.142 | 0.251 |
| Aspiration | Bad Bunny | APS | 0.433 | 0.023 | 8 | 0.365 | 0.501 |
| Deletion | Bad Bunny | APS | 0.37 | 0.022 | 8 | 0.304 | 0.437 |
| Maintenance | J Balvin | Speech | 0.808 | 0.011 | 8 | 0.774 | 0.842 |
| Aspiration | J Balvin | Speech | 0.075 | 0.008 | 8 | 0.052 | 0.098 |
| Deletion | J Balvin | Speech | 0.116 | 0.009 | 8 | 0.089 | 0.144 |
| Maintenance | J Balvin | APS | 0.5 | 0.029 | 8 | 0.414 | 0.586 |
| Aspiration | J Balvin | APS | 0.013 | 0.007 | 8 | −0.007 | 0.033 |
| Deletion | J Balvin | APS | 0.487 | 0.029 | 8 | 0.4 | 0.573 |

Pairwise contrasts of model probabilities were then calculated, comparing Speech to APS for each Artist and each Realization type. See Table 14, which shows how overall, they each change differently when they move from Speech to APS. Like Tables 10 and 11, which show the change between Speech and APS indicated with arrows, this model shows the estimate column as positive and negative changes for each Realization (positive movement in yellow). Each artist is moving in opposite directions from each other in their Realizations as they move from Speech to APS. There is no overlap in the upper and lower CL; therefore, the estimate is a strong indicator of the artists meeting in the middle.

**Table 14.** Contrasts comparing Speech and APS by Artist and Realization category.

| Contrast | Artist | Realization | Estimate | SE | df | Lower CL | Upper CL |
|---|---|---|---|---|---|---|---|
| Speech—APS | Bad Bunny | Maintenance | −0.159 | 0.019 | 8 | −0.203 | −0.115 |
| Speech—APS | J Balvin | Maintenance | 0.308 | 0.031 | 8 | 0.237 | 0.38 |
| Speech—APS | Bad Bunny | Aspiration | −0.408 | 0.023 | 8 | −0.461 | −0.355 |
| Speech—APS | J Balvin | Aspiration | 0.062 | 0.01 | 8 | 0.039 | 0.085 |
| Speech—APS | Bad Bunny | Deletion | 0.567 | 0.023 | 8 | 0.513 | 0.62 |
| Speech—APS | J Balvin | Deletion | −0.37 | 0.03 | 8 | −0.44 | −0.301 |

Table 15 and Figure 5 show the difference in percentage for each Realization and Artist, calculated as "APS %—Speech %". In the graph, confidence intervals for each are shown as shaded blue bars, and red arrows indicate the distance necessary between categories to conclude that those categories are significantly different from one another. For example, each time Bad Bunny has a negative change (APS percentage is less than Speech percentage for a certain Realization), J Balvin has a positive change (APS percentage is greater than Speech percentage) in the same category, and vice versa. The red arrows indicate that all the positive vs. negative comparisons are significant, since no red arrows overlap within one Realization.

**Table 15.** Contrast between APS % and Speech % per Realization and Artist.

| Contrast | Estimate | SE | df | Lower CL | Upper CL |
|---|---|---|---|---|---|
| Bad Bunny Maintenance | 15.87% | 0.0190 | 8 | 9.27% | 22.46% |
| J Balvin Maintenance | −30.84% | 0.0309 | 8 | −41.55% | −20.13% |
| Bad Bunny Aspiration | 40.80% | 0.0231 | 8 | 32.79% | 48.81% |
| J Balvin Aspiration | −6.19% | 0.0100 | 8 | −9.67% | −2.71% |
| Bad Bunny Deletion | −56.67% | 0.0232 | 8 | −64.71% | −48.63% |
| J Balvin Deletion | 37.03% | 0.0302 | 8 | 26.57% | 47.50% |

Confidence level used: 0.95. Confidence level adjustment: Šidák's method for 6 estimates.

Artist and Realization

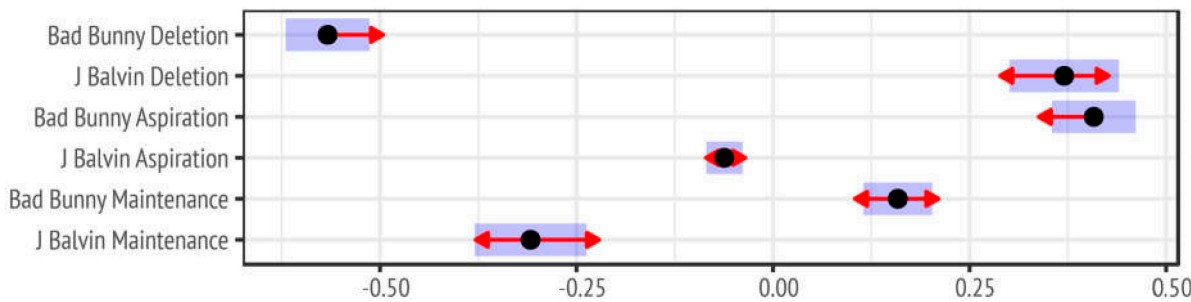

APS—Speech Difference in Percentage

**Figure 5.** Comparison of Realization percentages between APS and Speech.

Table 16 and Figure 6 are a comparison of Bad Bunny and J Balvin within Performance Mode and Realization type, which are contrasts calculated by comparing Bad Bunny to J Balvin within one Mode and one Realization type. For example, the difference between Bad Bunny and J Balvin is less pronounced in the deletion category when moving from Speech to APS (a difference of 82% moving to something much closer to zero −11.65%). Similarly, the difference between Bad Bunny and J Balvin decreases when moving from Speech to APS in the maintenance category. The opposite result is seen in the aspiration category, where Bad Bunny and J Balvin were closer together in the percentage of time that they used this realization while speaking but moved farther apart in the percentage of time they used this Realization in APS.

**Table 16.** Comparison of Bad Bunny and J Balvin within Mode and Realization type.

| Contrast | Estimate | SE | df | Lower CL | Upper CL |
|---|---|---|---|---|---|
| APS Maintenance | −0.3033 | 0.03403 | 8 | −0.4213 | −0.18544 |
| Speech Maintenance | −0.7704 | 0.01263 | 8 | −0.8142 | −0.72668 |
| APS Aspiration | 0.4198 | 0.02360 | 8 | 0.3381 | 0.50157 |
| Speech Aspiration | −0.0501 | 0.00886 | 8 | −0.0808 | −0.01941 |
| APS Deletion | −0.1165 | 0.03626 | 8 | −0.2421 | 0.00917 |
| Speech Deletion | 0.8205 | 0.01166 | 8 | 0.7801 | 0.86092 |

Confidence level used: 0.95. Confidence level adjustment: Šidák's method for 6 estimates.

*4.3. What Motivations Might Cause a Difference in the Pronunciation of Bad Bunny and J Balvin's APS and Spontaneous Speech?*

The creation of a pan-Latinx identity has always been a motivating factor behind reggaeton (Kattari 2009), and the fact that the Realization of coda /s/ in the artists' APS is moving away from the indexical pronunciation of their regional variants could be evidence that pop music is bridging the variants via dialect convergence (see Chambers 2002) and

leading to dialect leveling (see Siegel 1985) in APS. Previously, dialect leveling in Spanish language pop music tended towards coda /s/-maintaining variants, whereas the data in the present study show movement towards variants with coda /s/ reductions that were historically stigmatized. Bad Bunny and J Balvin are shifting their coda /s/ pronunciation from their spontaneous speech to their APS in opposite directions, with both artists' coda /s/ realizations converging towards a middle ground.

Performance Mode and Realization

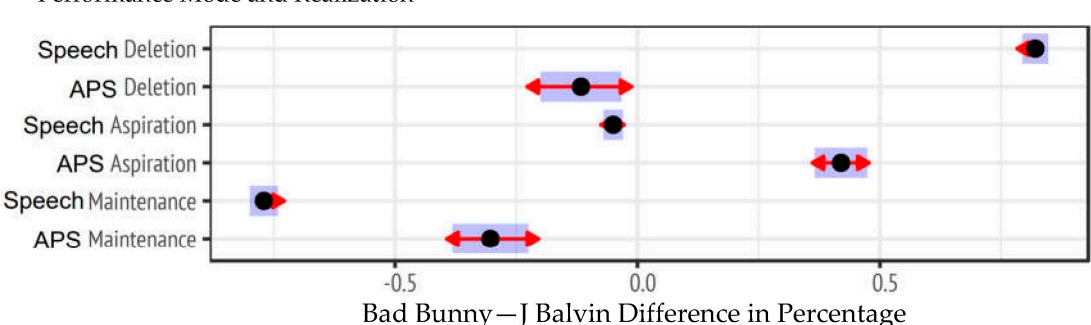

**Figure 6.** Comparison of Realization percentages between Bad Bunny and J Balvin.

This convergence could be facilitated by how easy it now is to connect with others online, watch others on social media, on TikTok videos, and on YouTube, and how new words, slang, and pronunciations are able to diffuse more quickly than ever before. This can be observed when a word specific to a regional variant is used in a song, and then it diffuses throughout the Spanish-speaking world. The term *tusa* 'heartbreak,' was a term used in Colombia, but because it was used in a song of the same name by Karol G, it is now widely used, in a sense coalescing those who use the term around a very specific identity. The language and pronunciation used in popular music are especially relevant to US Latinx who are not in the Latin American homeland of their ancestors. They tend to seek out Latinx culture via popular culture such as Latin music as a link to defining their identity. These generations of US Latinx are potentially the most impacted by the coda /s/ pronunciation that they consume via Latin music.

This convergence could be tied into the concept of glocality (Manca et al. 2021), where the formation of an identity can use pronunciation as the puzzle piece that sets it apart from one group and that connects it with another. Reggaeton songs have long attempted to bridge the divides between internal divisions of nationalities within Latin America, with songs that call out all the countries, such as the 2004 hit, "Oye Mi Canto" (N.O.R.E 2023), and the convergence in pronunciation could be a linguistic step in the same direction, where a pan-Latinx pronunciation can evolve that people around the Spanish-speaking world can sing along to, understand, and identify with.

A competing theory is that this new Caribbean-leaning pronunciation is becoming the model for Latin pop music pronunciation, much like Trudgill (1983) describes American pronunciation becoming the model for British pop music starting in the late 1950s. This theory can also claim some support by noting artists singing with markedly indexical Caribbean phonology who do not typically use that pronunciation in their spontaneous speech, such as (Mexican ancestry) Becky G and (Spanish artist) Rosalía. Lending more weight to this theory is the rising star of a new generation of Regional Mexican music, the Mexican-born Peso Pluma (Barrientos 2023; Sayre 2023). In the song "Plebada", he uses deletions of coda /s/, which are even noted in the lyrics with apostrophes: "problema'", "cargao' lleno'", (Genius Lyrics 2023), but if you listen to his speech in an interview (Al Rojo Vivo 2023), you can note his sibilant [s] pronunciation.

## 5. Conclusions

This mixed-methods study looked in detail at the artistic performance of identity via the sociophonetics (coda /s/) of the two top Latin music artists in the global market, Bad

Bunny and J Balvin. The corpus was composed of spontaneous speech found in interviews and artistic performance speech (APS) found in top songs on the Billboard charts. Coda /s/ pronunciation was chosen because it is a very salient feature in the artists' singing and speech, which is indexical of their identity.

Bad Bunny overwhelmingly used deletions in his spontaneous speech—which is typical of an island Puerto Rican—but he used a statistically significant amount of maintenance and aspiration in his APS. J Balvin primarily used maintenance in his spontaneous speech—which is typical of Medellín, Colombia—but he used about 50/50 maintenance and deletions in his APS. They are both shifting to a different pronunciation in their APS and converging towards a middle ground, and the difference is statistically significant. This could be the beginning of an identity-based pan-Latinx dialect levelling that is, on the one hand, the "in-crowd" pronunciation with covert prestige but, on the other hand, is part of the formation of an evolving multi-regional connector variant diffused through popular music and pop culture.

**Funding:** This research received no external funding.

**Institutional Review Board Statement:** Not applicable.

**Informed Consent Statement:** Not applicable.

**Data Availability Statement:** Data available in a publicly accessible repository that does not issue DOIs. This data can be found here: https://github.com/naranjohayes/BadBunnyJBalvinCodaS (accessed on 10 July 2023).

**Conflicts of Interest:** The author declares no conflict of interest.

## Appendix A

**Table A1.** Realization by Artist(s) in APS by song.

| APS | Realization | Bad Bunny | | J Balvin | | Totals |
|---|---|---|---|---|---|---|
| Cardi B, Bad Bunny, J Balvin—"I Like It" | Maintenance | 2 | 9% | 1 | 7% | 3 |
| | Aspiration | **15** | **68%** | 0 | 0% | 15 |
| | Deletion | 5 | 23% | **13** | **93%** | 18 |
| | Totals | 22 | 100% | 14 | 100% | 36 |
| J Balvin, Bad Bunny —"La canción" | Maintenance | 4 | 9% | 17 | 45% | 21 |
| | Aspiration | **32** | **71%** | 2 | 5% | 34 |
| | Deletion | 9 | 20% | **19** | **50%** | 28 |
| | Totals | 45 | 100% | 38 | 100% | 83 |
| J. Balvin, Bad Bunny —"Qué pretendes" | Maintenance | **13** | **37%** | **40** | **85%** | 53 |
| | Aspiration | 9 | 26% | 0 | 0% | 9 |
| | Deletion | **13** | **37%** | 7 | 15% | 20 |
| | Totals | 35 | 100% | 47 | 100% | 82 |
| Jhay Cortez, J. Balvin, Bad Bunny —"No me conoce (Remix)" | Maintenance | 5 | 18% | **16** | **80%** | 21 |
| | Aspiration | 9 | 32% | 0 | 0% | 9 |
| | Deletion | **14** | **50%** | 4 | 20% | 18 |
| | Totals | 28 | 100% | 20 | 100% | 48 |

**Table A1.** *Cont.*

| APS | Realization | Bad Bunny | | J Balvin | | Totals |
|---|---|---|---|---|---|---|
| J. Balvin, Dua Lipa, Bad Bunny, Tainy —"Un día (One Day)" | Maintenance | 2 | 13% | **15** | **68%** | 17 |
| | Aspiration | 4 | 25% | 0 | 0% | 4 |
| | Deletion | **10** | **63%** | 7 | 32% | 17 |
| | Totals | 16 | 100% | 22 | 100% | 38 |

| APS | Realization | Bad Bunny | |
|---|---|---|---|
| Casper, Nio García, Darell, Nicky Jam, Bad Bunny, Ozuna—"Te boté, Remix" | Maintenance | 1 | 6% |
| | Aspiration | **9** | **53%** |
| | Deletion | 7 | 41% |
| | Totals | 17 | 100% |
| Bad Bunny, Drake—"Mía" | Maintenance | 17 | 29% |
| | Aspiration | **27** | **46%** |
| | Deletion | 15 | 25% |
| | Totals | 59 | 100% |
| Bad Bunny—"Vete" | Maintenance | 25 | 30% |
| | Aspiration | **35** | **42%** |
| | Deletion | 23 | 28% |
| | Totals | 83 | 100% |
| Bad Bunny—"Callaíta" | Maintenance | 10 | 26% |
| | Aspiration | 13 | 34% |
| | Deletion | **15** | **39%** |
| | Totals | 38 | 100% |
| Bad Bunny, Jhay Cortez —"Dákiti" | Maintenance | 1 | 2% |
| | Aspiration | **33** | **52%** |
| | Deletion | 29 | 46% |
| | Totals | 63 | 100% |
| Bad Bunny —"Si veo a tu mamá" | Maintenance | 10 | 33% |
| | Aspiration | **11** | **37%** |
| | Deletion | 9 | 30% |
| | Totals | 30 | 100% |

| APS | Realization | J Balvin | |
|---|---|---|---|
| Nicky Jam, J. Balvin —"X (Equis)" | Maintenance | 5 | 19% |
| | Deletion | **21** | **81%** |
| | Totals | 26 | 100% |
| Anuel AA, Daddy Yankee, Karol G, Ozuna, J Balvin —"China" | Maintenance | **9** | **53%** |
| | Aspiration | 1 | 6% |
| | Deletion | 7 | 41% |
| | Totals | 17 | 100% |
| Ozuna—"Baila baila baila (Remix)" Ft. Daddy Yankee, J Balvin, Farruko, Anuel AA | Maintenance | **9** | **64%** |
| | Deletion | 5 | 36% |
| | Totals | 14 | 100% |

**Table A1.** *Cont.*

| APS | Realization | Bad Bunny | | J Balvin | Totals |
|---|---|---|---|---|---|
| Black Eyed Peas, J Balvin —"Ritmo (Bad Boys for Life)" | Maintenance | 7 | 18% | | |
| | Deletion | **31** | **82%** | | |
| | Totals | 38 | 100% | | |
| DJ Snake, J. Balvin, Tyga —"Loco contigo" | Maintenance | **20** | **51%** | | |
| | Deletion | 19 | 49% | | |
| | Totals | 39 | 100% | | |
| Sech, Daddy Yankee, J Balvin, Rosalía, Farruko —"Relación, Remix" | Maintenance | 11 | 44% | | |
| | Deletion | **14** | **56%** | | |
| | Totals | 25 | 100% | | |

The highest number and percentage of APS realizations per artist in each section are bolded.

## Notes

[1] Regarding terminology, there does not appear to be a fitting term to contrast with "spontaneous speech" that adequately describes the linguistic features of scripted artistic performance, therefore the term *artistic performance speech* (APS) will be used in this study.

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
