# Peer review of "Meeting in the Middle: Sociophonetic Convergence of Bad Bunny and J Balvin’s Coda /s/ in Their Artistic Performance Speech"

_languages, doi:10.3390/languages8040287_

Round 1

Reviewer 1 Report

Comments and Suggestions for Authors

The author studies the phonetic realization of coda /s/ - a reliable shibboleth in Spanish dialects according to previous literature - in the sung productions of two Latin music artists: Bad Bunny and J Balvin. Both artists seem to change their pronunciation when they sing and tend to use a pan-Latinx accent that suggests some word of dialect-levelling.

While the article is unquestionably very interesting for sociophoneticians and people studying the linguistics of the Spanish-speaking world, there are quite a few issues. In particular, some details are absent from the methodology section, and the results section is very confusing. I’ve made a number of suggestions below in the hope that they can help the author improve the strength of their article. Roughly speaking: more definitions are needed, a more accurate and more comprehensive description of the methods is needed, and the results portion should be made much clearer. As a secondary remark: authoritative references from general sociolinguistics and sociophonetics (not just Spanish-language sociolinguistics) are missing: no established reference for dialect-levelling, no reference for the computation of the center of gravity (which clearly does not go without saying), etc.

A quick definition of what reggaeton music is would be welcome. As someone who lives in Europe, who knows next to nothing about Spanish-language sociophonetics, but who has been a sociophonetician for over 20 years, and who has been involved in quite a few projects about the singing voice, I am sorry to say that I have never heard the term…

I wasn’t able to find a justification for the time span (2018-2020) that the author has chosen. What’s the reason for not including 2021-2022 at least?

The abstract quite aptly summarizes the context, goal, and main results of the article. However, some non-essential details should perhaps be left out at this stage. For instance: that the tokens were labelled with Praat (and that Praat is open source software), and that statistical analysis was performed with SPSS is not crucial at this point. And also “Is the difference statistically significant?” (it appears in the Introduction as well) is not a question you normally ask in an abstract or an Introduction. I have the impression that most empirical studies in linguistics and sociophonetics rely on data and some sort of tool to assess the reliability of the findings… these tools very often come from the field of statistical inference, so mentioning this in the abstract is perhaps secondary, and I’d say that, to a certain extent, it comes down to stating the obvious.

Line 81: maybe the first occurrence of “center of gravity” should be made more explicit by adding “spectral center of gravity”: while it’s easy to understand what “duration” means in this context, “center of gravity” may not be clear for people who have never measured the spectra of fricatives. And then in the next line, “in milliseconds” and “in Hertz” can probably be removed.

And also, line 82, it would be useful if the author explained what a lower center of gravity means in terms of the three realizations – with just a few more definition and clarifications, this article could target a wider readership, including researchers who are interested in the singing voice but know very little about Spanish sociophonetics or sociolinguists who know nothing about acoustic phonetics.

“Terrell (1977) noted that there is some evidence that the use of aspiration and deletion may be more favored by Puerto Rican males, younger generations…” How old are Terrell’s younger generations now? Has the trend evolved in the same directions over the last 45 years or so?

"non-maintenance of coda /s/ such as aspiration and deletion": this is slightly ambiguous. Technically, one could argue that /s/, whose most frequent allophone is [s], is either maintained (produced as another allophone: [h]) or deleted. So it would be more accurate to say something like: coda /s/ is either realized as [h] or deleted. And also, “aspiration” is generally and ill-defined term in phonetics. I strongly encourage the author to define it in acoustic and articulatory terms as early as possible in the article. I was able to make out that aspiration meant glottal frication thanks to the reference to Hualde (line 70-71) but it would be clearer if the concept were formally defined early in the article.

L102: “but very minimal tokens of glottal stops were found in the data”: not all variants are equally salient: is it really the case that glottal stops were rare or could it be that the recording/signal processing/mixing/mastering processes made glottal stops harder to detect?

L103: “Now we will consider coda /s/ as it pertains to Colombian Spanish regional variants, which are relevant to the artist, J Balvin.”: this type of transition is useful in academic theses and dissertations, but it can probably be left out in a journal article.

L113: “Circled in red” à “circled in red”

L119-120: “which we will discuss in Language Attitudes later this section.” à there’s something unexpected in this sentence (syntax?).

L120-121: “Next, we consider what makes pronunciation indicative of in-group identity.”: ditto; it is appropriate for a thesis, but journals generally make do without such transitions.

L123-127: the reference to Mack (conference proceedings with a paper focusing on /s/ in Puerto Rican Spanish) is not totally adequate here: it is used by the author to define indexicality in general, and not specifically indexicality as applied to the dialects and accents of Spanish. One or more authoritative references should be used here (like Reed, further below, or more established researchers in sociolinguistics, e.g N. Coupland,  J. Stuart-Smith, etc.). And also: Mack seems to be very relevant to the current topic, and, quite unexpectedly, the author did not review Mack’s findings…

L130: I was thinking that perhaps a more telling example could be used here instead of the American Southern drawl because unless the reader is a native speaker of American English, he or she will probably not be able to understand what this specific “drawl” (which is an impressionistic word) refers to. Whereas an illustration in any language with appropriate phonetic symbols and universal phonetic terminology would certainly be more intelligible.

L139-140: “In the same way, someone might infer that my consistent maintenance of coda /s/ and lack of deletions in Spanish are indexical of my Mexican ancestry, which would be a correct observation.”: this is very unclear: in the absence of double quotes, my interpretation is that the author is providing autobiographical information here. While this would be fine in a conference presentation, I tend to think this is not really appropriate here and there are dozens of potential illustrations for the meaning of “indexicality” that would spare the author the need to resort to autobiographical data.

L141: “Indexicality leads us to a discussion of language attitudes.”: transition not necessary.

L180: I am not totally convinced that the citation from Bad Bunny’s interview is necessary. Here I’d rather have two or three other references to studies where attitudes were scientifically analysed rather than the artist’s view on discrimination in a context that is highly constrained by events that involved another social group.

L185: “In the face of such negativity and discrimination, why continue to speak in a way that is so looked down upon? This leads us to covert prestige.”: yet another unnecessary transition. I feel that the content of such transitions could be rephrased and appear as introductions to the following sections.

L187: “Covert Prestige of Stigmatized Variants” à the title sounds very much like a pleonasm: isn’t covert prestige always about using non-standard/stigmatized variants?

L 188: “If copula absence is an iconic feature of African American English, then one could agree that aspiration and deletion of coda /s/ are iconic features of Caribbean Spanish.” à I don’t think the reference to AAE is necessary: not everybody is familiar with AAE or with the English language, so the comparison will be useless for many readers. I’d start straight away with the raw facts, ie., “Aspiration and delection of coda /s/ are…”.

L229: “which leads us to this study: the variable pronunciation of coda /s/ in Spanish especially as it pertains to music”: same recurring comment: unnecessary transition in journal style.

L235: “Owing to the prohibitive nature of their incredible popularity, no personal interviews were able to be conducted.”: I think it goes without saying.

L247: “Biography of Benito” and l264: “Biography of Jose”: I’d keep their artist’s names for section titles so that readers can more easily navigate through the article.

The number of /s/ tokens appears in Table 5. However, I think it would be useful to have this type of quantitative information (how many tokens, total duration of songs and interviews, etc.) as early as Sections 3.21 and 3.22. And at one point in Section 3.21, the list of selected songs is necessary. Perhaps a table with song title, song duration, and number of /s/ tokens is in order.

More generally, the author is welcome to be as accurate as possible regarding data selection and collection: bear in mind that everything in Section 3 should be explained with full details so that other researchers can replicate the study.

L318: just a quick remark: the use of Word files is rather unexpected; it is generally more common to work with plain text files especially when one is planning to perform automatic searches, and process text files programmatically.

L327: Is the diacritic below the [s] supposed to represent voicelessness? If so, then it should be horizontally aligned with the [s] but… [s] being voiceless, I don’t think the diacritic is necessary.

P10: Is the concept of trace-mark defined in Hualde (2014)? If so, the author should make it clear that the concept comes from this study. If not, references are needed. And footnotes 2 and 3 should probably appear in the body of the text.  

Just a thought: could it be that /s/ coded as trace-marked are just tokens exhibiting weak [h]?

In the caption of Figure 4, I think “Unmarked Deletion” should be removed.

I suppose Figures 3 and 4 come from interviews, not songs. Please make this explicit.

L355: “I also took acoustic measurements to verify the categorical labelling”: please say which acoustic measurements were made.

L357: “Many other variables were coded which were not included in the study.”: if you don’t include them, why mention them at all?

L360: “several statistical tests”, l362: “Several chi-squared tests”: “several” is very vague. This part is rather uninformative. The author should either say which test they used and how many of these were used. In this context, using “statistical tests” to check for “statistical significance” is a little redundant. I suppose 5 lines or so could be discarded here and the paragraph would start with “A multinomial logistic regression was…” which is exactly the type of information the reader is looking for. Also: why use capitalization for “Mode of Communication” and not for “artist” when both terms represent independent variables. If the author chooses to capitalize factors (which is common practice in statistics) then they should use an initial capital for “artist” too in the context of statistical tests.

L373: “This chapter discusses the results”: I’d call this a “section”, not a “chapter”.

L376: “What are the variants found in the pronunciation of coda /s/ in the top songs versus in the spontaneous speech of the two top global music artists, Bad Bunny, and J Balvin?”: a little hard to parse: please make it more simple. And why start using “top songs” when readers are at this point in the article used to “APS”?

L378: looking at the chi-square tests: why should the degrees of freedom between Bad Bunny (4) and J Balvin (3) be different? If I understand correctly, one chi-square test was performed for each artist to check whether the number of tokens in each of the four possible classes (unmarked deletion, trace-marked deletion, aspiration, and maintenance) were independent of mode of communication. If this is the case, then degrees of freedom should be identical for the two artists.

L 389: Table 5: personal preference: I’d rather have a bar plot instead of the table.

Starting from line 409: I am not familiar with the concept of direction of movement. So it’s either I am not aware of very recent developments or the terminology is not the one I am used to. In any case, I’m afraid many readers won’t be familiar with the concept and would ask for clarification if they could. Please explain what this means.

L444: “Bad Bunny had a statistically significant relationship between aspiration, and of trace-marked deletion of coda /s/ and his APS Mode of Communication at p < .001.”: hard to parse.

It is unclear which threshold for p-values the author uses. They say that p = .016 is significant, so we can infer that the threshold is 0.05. But then on line 447:  “approaching statistical significance at p < .093”… the very basic principle underlying Fisherian null hypothesis testing is that the decision should be dichotomous: it’s either significant or not significant at all.

L554: Duration in APS is very sensitive to tempo. It’s actually also sensitive to speech rate in interviews. It would probably be more accurate to express the duration of coda /s/ as a proportion of the duration of the whole word in order to provide some sort of speech rate normalization. Otherwise, it may well be that one artists speaks faster than the other, in which case, the observed duration difference could simply be due speech rate. And also (although I am not really familiar with these artists): it may well be that one of them favours high tempos and shorter note durations.

L567: “were the shortest duration in milliseconds.”: there’s no need to repeat “in milliseconds” (and this applies to other occurrences elsewhere in the text). When I was a student, I was taught that units should be explicitly mentioned. But in the case of duration, whether you’re using milliseconds or seconds doesn’t make any difference.

L578: “Is the difference statistically significant?”. Again, this is not a sociophonetic question. The question is: is there a difference. Then, in order to answer the question, researchers use the concept of statistical significance. But it’s just a tool to assess the robustness of a difference, and the reliability of this tool should not be overstated. In other words, statistical differences are assessed in a Results section in scientific articles, but scientists don’t generally explicitly ask the question, and it is clearly not the kind of question you want to use as a section title.

L584: I am having a hard time interpreting Table 10 (BTW, I’ve taught statistics to graduate students and Phd candidates in linguistics for about 20 years). First I may be wrong, but the use of “realization” is very ambiguous. Because here, it seems to be used as a synonym for “maintenance”. While on other occasions “realization” can be interpreted as the type of realization (unmarked, trace-marked, [h], maintenance). Then it’s hard to make sense of the “relationships between variables”. Then “center of gravity” (COG) is used as a variable but we have no means of knowing how this was computed. Bear in mind that power spectra are very sensitive to background noise (or music), digital sound processing techniques (like compression), etc. So, not only is COG not very robust but it is also almost useless in this article if the author doesn’t clearly explain how it was computed, and doesn’t use authoritative references (apart from Erker) for the measurement of COG. “Word boundary issues” is listed as a variable in the table but it was never mentioned before.

L592: dialect levelling: you need to cite works by people who developed the concept

Overall regarding statistics, it should be made much clearer which results correspond the chi-square tests and which to the multinomial logistic regression. The Results section is very confusing.

“I continue to track its evolution in my longitudinal research”: the tone of this sentence is unexpected in a journal article.

I’d definitely add a sort of disclaimer about the shortcomings of using commercial recordings to assess pronunciation as i) digital signal processing (DSP) techniques alter the voice signal and its phonetic content, and they may alter it differently depending on the artist (not because of the artist, but because of the specific studio each one uses): in other words, suppose Bad Bunny works with a team for the mixing and mastering of his songs, and J Balvin works with another team for the mixing and mastering, then “team and their DSP habits” rather than “artist” may be the relevant factor; ii) background noise and instruments will alter the acoustics of the singers’ voices and clearly bias COG measurements; iii) it may be that some variants are more preserved than others in studio recordings so an auditory analysis might well lead to missing variants that have been made less audible during mixing and mastering: for examples, de-essing (removing the sibilance of fricatives) is a very common technique in DSP applied to the voice, and by definition, it strongly alters COG, and probably impacts the perception of [s].

Comments on the Quality of English Language

Overall quality of English language is satisfactory. A limited number of sentences were hard to parse. A very limited number of prepositions may not be the default solution a native speaker would come up with (but this may be due to variation between Englishes). But these are very minor issues.

Reviewer 2 Report

Comments and Suggestions for Authors

My recommendation is for the author to revise and resubmit this contribution for reconsideration. Although I see the originality of comparing the two Latin Pop artists’ use of /s/, there are a number of areas where improvements need to be made for this manuscript to be publishable. Below are sections dealing with major concerns, details, and references.

Major comments

Categorical dependent variable – There are a number of issues that need to be addressed here:

  • The author makes a distinction between an “Unmarked Deletion” and a “Trace-Marked Deletion”. I have not seen any literature that attempts to make such a distinction, despite how extensively this variable has been studied over the past half century. While the description on lines 331-350 is imprecise (“empty spot” or “trace effect”), from what I can tell this distinction seems to be based on a misinterpretation of the acoustic information presented in the spectrogram. Specifically, the “Trace-Marked Deletion” would appear merely to be a deletion prior to a stop or affricate consonant while an “Unmarked Deletion” would appear to be a deletion prior to a continuant consonant. Part of what makes me believe this to be the case is how TextGrid tier 1 in Figure 4 is delimited. There is an unlabeled interval between the first two syllables of artistas. This interval corresponds to near silence in the signal that is really associated with the closure for the first /t/. By the same token, what is labeled “[deletion]” is really just the closure associated with the second /t/. It is empirically challenging to determine that the closure of the second /t/ is longer because the /s/ was deleted. Consequently, unless solid motivation can be provided for making this distinction, I recommend collapsing the two deletion categories into one and removing all references to said distinction.
  • Particularly given the results of the study, I think that the use of categorical coding is validated in this case. However, the author should take into consideration the discussion in Gradoville, Brown, and File-Muriel (2022) about some of the issues with categorical coding, particularly with variants that are between aspirated and deleted, where lower inter-rater reliability was found in impressionistic coding. While the author coded based on visual inspection of the spectrogram, the author would have had to make judgments that in some cases might not have been clear. Did the author carry out any sort of inter-rater or intra-rater reliability process? If not, it would be advisable to do so.
  • The author describes recognizing that the sibilant variant may be voiced, but makes no such recognition regarding aspiration. Note that File-Muriel and Brown (2010, 2011) measured voicing as a third acoustic measurement precisely because it cannot be guaranteed that aspiration is not accompanied by voicing, probably more properly considered breathy voice. How were such cases treated in the categorical coding?

Description of methodology and the methodology itself – The description of the methodology seems incomplete, disorganized, and inconsistent in some cases with what is presented in the results section. Notable issues include:

  • Section 3.5 (lines 352-358) is called “Independent variables”, but it almost exclusively describes dependent variables. Only the last sentence seems to address independent variables, only to say that other unnamed variables were coded but not included. Given the unbalanced nature of the data set, some independent variables that address phonetic context are definitely necessary in order to be sure that the differences observed between two two speakers in different conditions are due to speaker or condition and not that the contexts are different. Overall rates only tell part of the story.
  • Related to the previous point, in Table 4.4 in the results section, a word class variable is randomly presented that was not described in the methodology. The methodology and results section need to be in sync.
  • On lines 355-357, the author describes taking acoustic measurements in order to “verify the categorical coding”. What measurements were taken? How were they taken? What was the process to verify the categorical coding? Keep in mind that something like center of gravity, which is described elsewhere in the manuscript, is affected by the non-speech sounds in the acoustic symbol, so what was done in cases when the supposed /s/ overlapped non-speech sounds?

Statistical methodology – The issue of statistical methodology should receive special attention for a number of reasons:

  • The description of the statistical methodology (lines 359-370) is scattered and vague. There is discussion of “several” chi-squared tests and “several” measures of association. What were the chi-squared tests used on? What sorts of measures of association were used?
  • One lines 365-367, the author describes using a multinomial logistic regression. This definitely seems like the appropriate model to use for the data, but I find no mention of this regression model anywhere afterward. Moreover, there is no discussion about how model selection was carried out.
  • The author seems to make use of numerous chi-squared tests throughout the manuscript, although it is generally unclear in the results section what variables form the contingency table on which the chi-squared tests are based at any given time. It’s significant or it’s not, but it is not clear what is or is not significant. There is a more general issue, however, that pertains to the sheer number of chi-squared tests used. In addition to the fact that the chi-squared test can only deal with two variables at a time, the large number of chi-squared tests increases the probability of a Type I error. The more appropriate way to deal with this is to carry out a single unified multinomial logistic regression that accounts for all the variables and interactions at the same time.

Characterization of a “middle ground” or “center point of pronunciation” – Both historically and articulatorily, the process involving /s/ variants is usually regarded as [s] > [h] > 0. The author has characterized Bad Bunny using aspiration in performance (vs. deletion in interviews) and J Balvin using deletion in performance (vs. sibilance in interviews) as some sort of “middle ground” or “center point of pronunciation”. If the supposed vernacular variants of Bad Bunny and J Balvin are deletion and sibilance, respectively, the middle ground would be aspiration. Bad Bunny moves toward aspiration in the music, but J Balvin is doing something totally different. He’s completely skipping over aspiration and going to deletion, which is almost a sort of hypercorrection toward the lowest status variant. Something that the author may want to consider is that aspiration does not occur at any great frequency in any of the data produced by J Balvin. If we assume that aspiration is indeed not part of his vernacular, maybe he is incapable of performing aspiration and has to use deletion in his performances because it is the only other variant available to him. Deletion seems like an exaggeration if we are taking Bad Bunny as the norm. This is also not unprecedented if, for example, you examine Anitta’s production of Spanish speech sounds, which seems to contain similar exaggerations.

Results section – This section is difficult to follow. Create an appendix that lists variant frequencies in each song/interview so that it can be examined in full. The way the results section is presented seems very scattered. It is not clear if every song/interview is discussed. Moreover, different songs/interviews seem to pop up at random. I would remove most of this detailed discussion, talk about the main trends, and limit the details to cases where someone is clearly doing something different from the overall trend. In addressing the interviews, it would be advisable to keep in mind who the interviewer was. For example, on line 449, an interview with Don Francisco is discussed. Don Francisco is Chilean, and Chilean Spanish is known for /s/ reduction, so we might not expect much accommodation there.

Speaking rate issues – The author should keep in mind speaking rate issues. All things being equal, a speaker will use a more reduced variant at higher speaking rates. Could differences between Bad Bunny’s interviews and music be explained by speaking rate? J Balvin’s linguistic behavior is unlikely to be explained in this way.

Song style issues – The author should keep in mind the genre of the song, not just the overall genre of the singer. For example, on lines 518-520, the author discusses J Balvin’s variant use in two songs, “I like it”, which has minimal sibilants, and “Un día”, which has mostly sibilants. The author has attributed this to audience and identity issues, but the two songs are very different in style. “I like it” has a lot of rap while “Un día” has more singing. The difference in variant usage could very easily be explained by that fact alone.

Musical style – The author should keep in mind that, although Bad Bunny collaborates with reggaeton artists and is associated with that musical genre, he is known especially for coming to prominence as a Latin trap artist, which has a somewhat different sound. J Balvin is primarily known for reggaeton. However, musical style is a way in which these two artists aren’t really the same.

Detailed comments

Note: initial number(s) refer to line numbers

11 – avoid using abbreviations that haven’t been previously defined, unless said abbreviations are widely known

115-117 – Are there any published empirical studies of /s/ in Medellín? It’s probably not ideal to simply assume that because Medellín is a highlands variety it will behave like Bogotá with respect to /s/. I’m guessing that Montes Giraldo may have also placed Cali into this highlands region. Earl Brown and colleagues have studied /s/ in multiple data sets from Cali, and it has /s/ reduction in both initial and final positions (Brown, 2009a, 2009b; Brown & Brown, 2012; File-Muriel & Brown, 2010, 2011; possibly among others). Consequently, it is not clear that Medellín Spanish is immune from /s/ reduction. Moreover, it is possible that J Balvin’s extreme maintenance of sibilant /s/ in video-recorded interviews is due to the Bogotá-based norm in Colombian media, not J Balvin’s vernacular.

287-288 – The statement that Latin music is a collaborative genre is overly broad. The recent wave of mass collaboration can probably be best associated with reggaeton and related genres that have arisen within the Caribbean. Prior to the mainstream turn of reggaeton, collaborations were far more rare. As a matter of fact, two artists (Miguel Bosé, Reyli Barba) released duet albums (Bien acompañado, Papito, Papitwo) of songs that they had previously sung as solo artists. It is probably a fair judgment to say that some of the collaborations of previously known non-reggaeton artists with reggaeton artists (Reik, Laura Pausini, Maná, etc.) may be done so that the artist does not completely disappear in the wave of reggaetonization of Latin Pop.

291-300 – More information on the corpus is necessary. How many songs/interviews were used? Moreover, if not every song/interview meeting the parameters was used, how did the author(s) go about selecting a subset?

303 – The author should keep in mind that, while a WAV file is not lossy, that does not mean that the signal was not subject to lossy compression between the time it was recorded and when it was downloaded by the author. Once sound information is lost due to lossy compression, converting it back to a WAV will not recover said information. This is especially relevant for something like center of gravity, which the author reports measuring, although no such results are presented.

355 – What does MADD mean?

469 – What does “13 tokens” refer to? Is this total N or is it the number of tokens realized in this way. This issue applies throughout this paragraph and elsewhere.

554 – The section title includes “Center of Gravity”, but there is no discussion of any measurements. To be clear, I would be inclined to recommend completely removing this section, as long as something is done about the inter-rater and/or intra-rater reliability of the categorical dependent variable.

564 – This reference to Erker (2012) is uninterpretable. What was done precisely following Erker (2012)?

576 (Table 9) – This would probably be easier to digest as a box plot. It is not fully explained what the numbers between “Shortest” and “Longest” mean.

581-582 – It seems problematic to assume that Bad Bunny basically only has one variant in his repertoire (deletion) and that his use of aspiration in the music is somehow unfaithful to his particular regional variety. Would it not be more appropriate to consider that a range of variants may be used in a variety with said variants distributed differently depending on the social circumstances?

584 (Table 10) – What statistical tests were used here? What were they used to compare? Moreover, if you aren’t going to discuss the data in here, should this table be put into an appendix or left out entirely? Any time you place a table or figure within the text, explain what you want the reader to see in there.

625-626 – Chanel Terrero was born in Havana, Cuba, but moved to Spain at 3 years of age. She clearly has some Spanish family members, but does she also have Cuban family members? If she does have contact with Cubans, this is part of her identity too. The thing to keep in mind is that the Spanish school system, which she passed through, seems like a bit of a pressure cooker when it comes to enforcing normative Peninsular Spanish in formal situations.

628 (and elsewhere) – The author references this being a mixed-methods study, but I am not sure what the qualitative aspect of this study is.

References

Brown, Earl K. (2009a). A Usage-based Account of Syllable- and Word-final /s/ Reduction in Four Dialects of Spanish. Munich: Lincom Europa.

Brown, Earl K. (2009b). "The Relative Importance of Lexical Frequency in Syllable- and Word-Final /s/ Reduction in Cali, Colombia." In Selected Proceedings of the 11th Hispanic Linguistics Symposium, ed. by Joseph Collentine, Maryellen García, Barbara Lafford, and Francisco Marcos Marín, 165-178. Somerville, MA: Cascadilla Proceedings Project.

Brown, Earl K. & Esther L. Brown. (2012). "Syllable-final and syllable-initial /s/ reduction in Cali, Colombia: One variable or two?" Colombian Varieties of Spanish, ed. by Richard File-Muriel and Rafael Orozco, 89-106. Madrid: Iberoamericana.

File-Muriel, Richard & Earl K. Brown. (2010). "The gradient nature of /s/-lenition in Caleño Spanish." University of Pennsylvania Working Papers in Linguistics, 16(2): 46-55.

File-Muriel, Richard & Earl K. Brown. (2011). "The gradient nature of s-lenition in Caleño Spanish." Language Variation and Change, 23: 223-243.

Gradoville, Michael S., Earl K. Brown, & Richard J. File-Muriel. (2022). "The phonetics of sociophonetics: Validating acoustic approaches to Spanish /s/." Journal of Phonetics, 91

Comments on the Quality of English Language

In general, the English used within the manuscript has minimal issues, although any subsequent version should be proofread by a native English speaker before resubmission. There are, however, issues that need to be addressed with the writing that don't pertain to the mechanics of English. I have commented elsewhere where I think some of these issues are, but there is a lack of clarity in a number of places and in other places the discussion seems to get lost in minutiae.

Round 2

Reviewer 1 Report

Comments and Suggestions for Authors

Most of my comments on the first version of the manuscript have been addressed in this new version.

I have only minor remaining comments.

Lines 139 to 145: I don’t understand why, now that examples of indexicality referring to the Spanish-speaking community have been given, the author provides us with an example of indexicality applied to the English-speaking world. My best guess is that this may be partly the result of my previous remark on the need for more “established” references. A solution would be to place Preston’s example before the Mack (2011) and Rosa (2010) examples.

Line 80: Again, Terrell is an old reference. It’s probably a good idea to insist on this by saying: “Terrell (1977) noted that in the late 1970s there was some evidence…”

Table 10: maybe more appropriate in the appendix?

Line 278 “in Table 5” shouldn’t be in italics.

Reviewer 2 Report

Comments and Suggestions for Authors

In general, this manuscript is substantially improved over the first version. Below are additional suggestions for improvement.

General comment with respect to references to songs: if I am not mistaken, songs should probably be referenced with quotation marks rather than italics. Moreover, when the song title in Spanish, only the first word of the title is capitalized unless the word would otherwise be capitalized anyway. English, on the other hand, usually capitalizes most content words in titles. References to songs within the text should be modified accordingly.

General comment on relative references to figures and tables: avoid deictics when referencing tables and figures; just refer to them by their name. This avoids circumstances where the location of a table or figure has to be modified, which causes the need to modify the deictic.

Lines 139-146 – maybe there’s a transition missing, but I’m not sure why there is an example here of an indexical feature in English.

Lines 148-156 – The discussion of attitudes of Spanish speakers in the United States needs to be more closely tied to the topic at hand. What bearing does it have on the usage of Bad Bunny and J Balvin?

Lines 309-317 – It probably would not be a bad idea to describe the criteria that were used for describing, based on the spectrogram, which variant had been produced. Ultimately, aspiration is a bit of a transitional variant: sometimes it has short, weak high-frequency frication noise, but sometimes the vowel formants just become more poorly defined.

Lines 320-321 (Figure 4) – I recommend showing some additional context after the last syllable so that the reader can see that there is indeed no fricative there.

Lines 325-326 (“Frication was noted in the spectrogram.”) -- This is stranded. It belongs in the discussion of the acoustic analysis above.

Lines 328-330 – Cite R and all relevant packages.

Lines 340-395 – This section is still hard to follow. Some of the analysis of the rationale for the use of variants seems speculative.

Line 441 – Table 16 does not show percentages.

Lines 476-480 – Keep in mind that Spanish-language music has had dialect leveling all along; in the past, it has been in the direction the tierras altas. What has changed here is that there is some movement toward the tierras bajas, whose variants within the larger Spanish-speaking community have historically had low status.

Lines 504-506 – I think that, in order to defend this position, you would want to see the spread of /s/ aspiration to non-”urban” Latin pop genres, particularly when the artists in question are not Caribbean. For example, I don’t believe that the singers of Miranda! use /s/ aspiration in their music, despite the fact that they are native speakers of a variety that is squarely in the aspiration camp. This also does not seem like something that Julieta Venegas or Jesse y Joy do, even if they collaborate with a reggaeton artist. If you want an example of an artist who seemed to shift his pronunciation toward /s/ reduction much earlier than this, compare Alejandro Sanz’s earlier works to what he did in “No es lo mismo” in 2004.

Lines 533-539 – I think this is probably your rationale for not using instrumental measurements, so it should probably appear in the methodology section.

Comments on the Quality of English Language

The English is again not an issue, but there are some things that should be carefully proofread.
